# E3 Ubiquitin Ligases: Key Regulators of TGFβ Signaling in Cancer Progression

**DOI:** 10.3390/ijms22020476

**Published:** 2021-01-06

**Authors:** Abhishek Sinha, Prasanna Vasudevan Iyengar, Peter ten Dijke

**Affiliations:** Department of Cell and Chemical Biology and Oncode Institute, Leiden University Medical Center, 2300 RC Leiden, The Netherlands; A.Sinha@lumc.nl (A.S.); P.V.Iyengar@lumc.nl (P.V.I.)

**Keywords:** cancer, E3 Ligase, PROTAC, signaling, SMAD, SMURF, TGFβ, tumor, ubiquitin

## Abstract

Transforming growth factor β (TGFβ) is a secreted growth and differentiation factor that influences vital cellular processes like proliferation, adhesion, motility, and apoptosis. Regulation of the TGFβ signaling pathway is of key importance to maintain tissue homeostasis. Perturbation of this signaling pathway has been implicated in a plethora of diseases, including cancer. The effect of TGFβ is dependent on cellular context, and TGFβ can perform both anti- and pro-oncogenic roles. TGFβ acts by binding to specific cell surface TGFβ type I and type II transmembrane receptors that are endowed with serine/threonine kinase activity. Upon ligand-induced receptor phosphorylation, SMAD proteins and other intracellular effectors become activated and mediate biological responses. The levels, localization, and function of TGFβ signaling mediators, regulators, and effectors are highly dynamic and regulated by a myriad of post-translational modifications. One such crucial modification is ubiquitination. The ubiquitin modification is also a mechanism by which crosstalk with other signaling pathways is achieved. Crucial effector components of the ubiquitination cascade include the very diverse family of E3 ubiquitin ligases. This review summarizes the diverse roles of E3 ligases that act on TGFβ receptor and intracellular signaling components. E3 ligases regulate TGFβ signaling both positively and negatively by regulating degradation of receptors and various signaling intermediates. We also highlight the function of E3 ligases in connection with TGFβ’s dual role during tumorigenesis. We conclude with a perspective on the emerging possibility of defining E3 ligases as drug targets and how they may be used to selectively target TGFβ-induced pro-oncogenic responses.

## 1. Introduction

Transforming growth factor-beta (TGFβ) was discovered more than thirty years ago as a secreted polypeptide from sarcoma virus-infected cells that promoted the soft agar independent growth of normal rat kidney (NRK) cells [1,2]. This biological property was associated with oncogenes, hence the name transforming growth factor. However, studies shortly thereafter discovered that this effect of TGFβ (unlike the effect of oncogenes) was reversible, and that TGFβ can have potent cell growth inhibitory effects [3]. Thirty-three human genes encoding distinct but closely structurally and functionally related TGFβ family members have been identified, which includes TGFβ1,-β2 and -β3, activins, inhibins, bone morphogenetic proteins (BMPs), myostatin, nodal and the mullerian inhibitory substance (MIS) [4,5,6]. The active TGFβ molecule is a dimer stabilized by hydrophobic interactions and a disulfide bond [7]. Following the binding of TGFβ to its specific receptors that are expressed on nearly all cell types, TGFβ regulates a plethora of biological processes, ranging from cell proliferation and differentiation, embryogenesis, hormonal synthesis and secretion, immunity to tissue remodeling and repair [6,8]. Because of its ability to induce cell growth inhibition and apoptosis in normal and pre-malignant cells, TGFβ has been described as a potent tumor suppressor [8,9]. In support of this notion, mutations in the components of the TGFβ signaling cascade have been identified in a number of human cancers, including hereditary nonpolyposis colon cancer, hepatocellular carcinoma (HCC), and pancreatic and ovarian cancer [10]. In contrast to this activity, TGFβ can also function as a tumor promoter by promoting cancer cell proliferation, stimulating epithelial-to-mesenchymal transition (EMT) and migration of cancer cells, and indirectly by acting on the tumor microenvironment, promoting angiogenesis, and/or immune evasion in advanced stages of tumor progression [8,9].

Active TGFβ induces the heteromeric complex formation of two single transmembrane serine/threonine kinase family receptors, i.e., TGFβ type I and type II receptor (TβRI and TβRII, respectively) [11,12]. Two molecules of TβRII associate with two TβRIs molecules thereby forming a tetrameric receptor complex [13,14]. Upon ligand-induced complex formation, the TβRII’s constitutively active kinase phosphorylates TβRI [also termed activin receptor-like kinase-5 (ALK5)] on serine and threonine residues in the juxta membrane glycine-serine residue-rich (GS) domain [15,16]. TβRI acts downstream of TβRII, and is the main component of the receptor complex that triggers various downstream signaling activities [17]. The signal is relayed from the receptors to the nucleus via cytoplasmic effector molecules, a process called SMAD and non-SMAD signaling [18,19].

The action of TGFβ is highly context dependent and is subject to positive and negative regulation in each step of the signaling pathway [6,20]. This is needed to enable spatio-temporal control and allow for crosstalk with other signaling pathways [21]. An important mechanism by which the expression, localization, and activity of TGFβ signaling components is controlled is by post-translational modification (PTMs). For example, PTMs that occur in receptors and SMAD proteins include phosphorylation, ubiquitination, sumoylation, ribosylation, and acetylation [22,23,24,25,26,27,28,29]. In particular, ubiquitination is emerging as a key regulatory mechanism by which intracellular signaling intensity, duration, and specificity is reversibly controlled through the action of multiple E3 ubiquitin ligases and deubiquitinating enzymes [30,31,32]. A concept is emerging that disruption of the ubiquitin modification and function of TGFβ pathway components may result in illicit protein activity leading towards human diseases including cancer.

The structure of the review is as follows. We first provide an overview of TGFβ intracellular signaling and describe how this is regulated after summarizing the biphasic role of TGFβ in cancer progression, we describe how ubiquitination regulates protein function and how TGFβ signaling is controlled by E3 Ubiquitin ligases. We specifically focus on the roles of individual E3 ligases in relation to their involvement in (dys)regulating TGFβ signaling in cancer progression. We provide a summary on strategies to selectively regulate specific E3 ligases and their substrates, which in the future may contribute towards a more effective and targeted treatment for TGFβ-induced cancer progression. Finally, we provide perspectives on future prospects of harnessing the E3 ligase network to target TGFβ signaling components. We do not discuss the role of deubiquitinating enzymes that impact TGFβ signaling but do refer the reader to excellent reviews on that topic [32,33,34].

## 2. SMAD and Non-SMAD Signaling

In the canonical SMAD-dependent pathways, the activated TβRI recruits and phosphorylates regulatory (R)-SMADs, i.e., SMAD2 and SMAD3, on their two C-terminal Serine residues (within the SxS motif) (Figure 1) [35,36]. Usually this TβRI-mediated Smad2/3 phosphorylation peaks around 45–60 min following receptor activation. The adapter protein named SMAD anchor for receptor activation (SARA) plays a role in the recruitment of these R-SMADs to the activated TβRI. Other members of the TGFβ family, such as the BMPs acting via BMP receptors, employ different R-SMADs as their effector proteins (SMAD1, 5, and 8) [37]. Once phosphorylated, SMAD complexes detach from their respective receptor and associate with the common partner SMAD4 also known as co-SMAD [38,39,40] (Figure 1), and these heteromeric SMAD complexes then translocate to the nucleus. Once inside the nucleus, these complexes can bind to specific promotors of target genes cooperatively with other DNA binding transcription factors to act as DNA sequence-specific transcriptional regulators of target genes [41]. Transcription coactivators also modulate nuclear SMAD complex transcription function. These type of co-activators include well characterized proteins i.e., CREB binding protein (CRE)/p300, p300/CBP-associated factor (P/CAF). SMAD3 and SMAD4, but not SMAD2 can directly recognize a specific DNA sequence (5′-CAGAC-3′) in promoter regions, termed the SMAD binding element (SBE) [42]. Inhibitory(I)-SMAD proteins, i.e., SMAD6 and SMAD7, are negative feedback regulators of TGFβ/SMAD signaling [43,44,45]. They regulate the intensity and duration of TGFβ family signaling responses. One mechanism by which SMAD7 inhibits TGF-β/SMAD signaling is by functioning as an adaptor to recruit the E3 ubiquitin ligase, SMAD ubiquitin regulatory factor 2 (SMURF2) to the activated TβRI, thereby targeting it for proteasomal degradation [27]. Whereas SMAD7 negatively regulates TGFβ, activin or BMP signaling, SMAD6 specifically inhibits BMP signaling. The stability of the interaction between SMAD7 with TβR1 is increased in the presence of serine-threonine kinase receptor-associated protein (STRAP), Yes-associated protein (YAP65), and atrophin 1-interacting protein 4 (AIP4) [46,47,48]. Moreover, inhibitory SMADs are regulated in expression and stability by other cues, and thereby enable cross talk with other signaling pathways [45].

It is important to note that besides the TβRI-initiated SMAD signaling pathway [49], non-SMAD (or SMAD independent) signaling mechanisms occur [50]. TGFβ can activate stress-activated kinase pathways, i.e., p38 and Jun N-terminal kinase (JNK) mitogen activated protein kinase (MAPK) pathways (Figure 1) [18]. These pathways when induced by TGFβ ligands, can signal independently, or cooperate with or fine-tune, the canonical SMAD signaling to trigger biological responses, such as EMT and apoptosis [51,52,53,54]. Another non-canonical signaling pathway for TGFβ occurs via the extracellular-signal-regulated kinases 1 and 2 (ERK1 and ERK2) MAPKs [55] (Figure 1). The rapid kinetics (peaking at 5–10 min post stimulation) of activation of ERK1 or ERK2 by TGFβ or BMP is similar to their activation by mitogenic growth factors. In addition, TGFβ has also been shown to affect the mTOR/phosphoinositide 3-kinase (PI3K)/AKT pathway [52,56]. Moreover, TGFβ can also activate Rho GTPases in certain cells, both directly downstream of TGFβ receptors or indirectly by inducing expression of exchange factors, and thereby affect cytoskeleton reorganization and cell invasion [57,58,59,60].

## 3. TGFβ as Tumor Suppressor and Tumor Promoter

In normal (healthy) and premalignant cells, the TGFβ signaling pathway prompts a tumor suppressive role. However, in advanced metastatic tumors, this pathway can be blunted or corrupted or even utilized by cancer cells to promote oncogenic functions [9,61] (Figure 2).

TGFβ induces cell cycle arrest in G1 phase by inducing the expression of cell cycle inhibitory proteins, like cyclin-dependent kinase inhibitors (CDKIs) p15^INK4B^ and/or p21^KIP1^, which in turn inhibit specific CDK functions [62,63]. TGFβ also represses the expression of growth inducing factors such as the oncogene *c-MYC*, and the ID family of transcription factors (*ID1*, *ID2*, and *ID3*), which also results in inhibition of cell proliferation [64,65,66,67]. Another anti-tumor function of TGFβ is its role in promoting apoptosis (Figure 2). In hepatocytes and B-lymphocytes, TGFβ promotes SMAD and the p38 MAPK-dependent transcriptional induction of the pro-apoptotic Bcl-2 family members BMF and BIM. These in turn induce the pro-apoptotic factor BAX, which leads to mitochondrial release of Cytochrome C and increase in caspase-dependent apoptosis [68,69]. Similarly, TGFβ signaling is known to repress BCL-_XL_ and anti-apoptotic BCL-2 family members leading to promotion of apoptosis [70,71,72]. In liver, TGFβ promotes the expression of pro-apoptotic protein, death-associated protein kinase (DAPK), in a SMAD dependent manner [73] (Figure 2).

In mice models, TGFβ has been found to regulate genomic stability [74,75,76] (Figure 2). Knockout of TβR1inhibited the phosphorylation of ataxia-telangiectasia mutated (ATM), p53, CHK2, and RAD17, which increased the radiation-sensitivity of cells derived from knockout mice compared to normal mice [77]. Similarly, SMAD4 conditional knockout mice with head and neck cancer, revealed a role for SMAD4 in the regulation of the Fanconi anemia/BRCA DNA repair pathway, also suggesting an involvement in genomic stability [78]. The same study demonstrated a correlation of SMAD4 knockdown with the downregulation of BRCA1 and RAD51 protein expression in head and neck squamous cell carcinoma (HNSCC) as well as increased expression of TβR1 and phospho-SMAD3 [78]. The exact mechanism of this regulation remains to be determined.

Mutation or functional inactivation of TGFβ receptors or downstream SMAD proteins have been associated with progression of malignancies [79], as it mitigates the TGFβ/SMAD-induced cytostatic responses, and leads to cell hyperproliferation. In particular, *TβRI*, *TβRII*, *SMAD2*, and *SMAD4* are frequently mutated, deleted, or attenuated (gene/loss of heterozygosity/expression) in certain cancer subtypes. Inactivating mutations or deletions of genes encoding TGFβ receptors and SMADs are common in esophageal, colorectal, and pancreatic adenocarcinomas. *SMAD4* inactivating mutations are common in gastric, colorectal, and pancreatic adenocarcinomas, whereas up to 20% of head and neck, bladder, cervical, and lung squamous carcinomas contain inactivating mutations in TGFβ signaling components like *SMAD2*, *SMAD3*, *SMAD4*, *TβR1*, or *TβRII* [8].

In late stages of cancer, TGFβ signaling can switch towards a tumor promoting function through diverse mechanisms (Figure 2). In the following section, we discuss the pro-oncogenic role of TGFβ signaling with a few examples. The following roles are specifically true for cases where SMADs or receptors are not functionally inactivated by mutations. The pro-oncogenic responses of TGFβ can be broadly classified into three major groups.

### 3.1. Epithelial-to-Mesenchymal Transition (EMT) and Invasion

EMT is an important mechanism by which epithelial cells acquire fibroblast-like properties. This is a characteristic feature in normal cells undergoing embryogenesis and wound healing, but also occurs in pathological processes like fibrogenesis and tumorigenesis. TGFβ is a well-known inducer of EMT (Figure 3). Notably, cells that overexpress SMAD7 or have reduced expression of SMAD3/4 show significantly decreased EMT in response to TGFβ [80,81]. TGFβ was found to induce the expression of transcription factors such as SNAIL, SLUG, TWIST, and FOXC3, which are of key importance to downregulate epithelial genes such as *E-CDH1* (encoding E-Cadherin) and mediating acquisition of mesenchymal features by cancer cells [80,82]. Acquiring mesenchymal characteristics is a prerequisite for the spread of cancer cells at secondary sites of the body and the process of metastasis. During breast cancer progression, TGFβ induces single cell migration through a SMAD-mediated pathway involving downstream activation of epidermal growth factor receptor (EGFR), Jun and Rho signaling pathways, and connective tissue growth factor (CTGF) [83,84]. TGFβ also stimulates the secretion of the parathyroid hormone related protein (PTHrP), which in turn induces the expression of bone homing receptor C-X-C chemokine receptor type 4 (CXCR4). CXCR4 promotes chemoattraction of the breast cancer cells to the bone secondary sites during metastasis [85,86] (Figure 2).

### 3.2. Promoting Angiogenesis

Angiogenesis is the process in which new vasculature is formed from existing blood vessels. This is central to the development of tumors as blood vessels provide a steady supply of blood and associated metabolites, cytokines, and growth factors, which promotes uncontrolled cell proliferation. TGFβ can promote angiogenesis through its effects on various angiogenic factors such as vascular endothelial growth factor (VEGF) and CTGF [85,87] (Figure 2). TGFβ also promotes the production and secretion of well-known angiogenesis promoting matrix metalloproteases, MMP-2 and MMP-9 [76,88]. It has been also shown that TβRI-mediated SMAD2/3 signaling promotes transcription of extracellular matrix proteins such as fibronectin and plasminogen activator inhibitor type 1 (PAI1), which regulate angiogenesis by inducing vessel maturation [89]. Thus, TGFβ can promote angiogenesis and cancer progression by promoting expression of a variety of angiogenic factors (Figure 2).

### 3.3. Immunomodulatory Effects

TGFβ has a broad immunomodulatory effect resulting in mostly pro-tumorigenic effects [90]. In short, the adaptive immune system, consisting of T and B cells, can eliminate tumor cells upon recognition. Through its immunomodulating role, TGFβ signaling can inhibit the function of antigen presenting cells, thereby decreasing T cell activation and decreasing elimination of tumor cells [91]. In addition, TGFβ directly inhibits CD4+ and CD8+ T cells, as well as natural killer (NK) cells. The effect of TGFβ on the innate immune system, like neutrophils and macrophages, also mostly results in pro-tumorigenic effects as it drives these cells from a type 1 differentiated cell into a most immature type 2 cell. This modulation occurs in neutrophils, macrophages, and in the T cells leading to enhanced release of TGFβ into the tumor microenvironment [74,92] (Figure 2).

Here we have discussed, with examples, the pro-tumorigenic effects of TGFβ though it is noteworthy that there are other ways (e.g., regulating stemness, relapse of cancer, etc.) through which this signaling can promote cancer. Overall, it can be concluded that tightly controlled TGFβ signaling is critical for normal functioning of the cell and that perturbations of this pathway can contribute to cancer progression.

## 4. Controlling Protein Function by Ubiquitination

The process of ubiquitination is required for controlling the stability, location, and function of proteins [93]. In this process, the target proteins are modified at lysine (K) residues through the covalent attachment of a small protein ubiquitin (8 KDa) (Figure 3A) [94,95]. Based on the number of ubiquitin molecules attached to a particular position, the ubiquitination can be either mono- or poly-ubiquitination. In the case of monoubiquitination, only a single ubiquitin molecule is attached to a particular lysine. Monoubiquitination has been shown to regulate intracellular transport, protein sorting, chromatin regulation, as well as rare cases of degradation [96,97,98,99]. Polyubiquitination is a crucial step for proteasomal and autophagic degradation of proteins but can also regulate protein function, for example by controlling DNA damage repair, intracellular signaling, and ribosomal biogenesis [100,101,102]. An additional step of complexity results from the different topologies of polyubiquitin chains conjugated through various lysine residues (e.g., K6, K11, K27, K29, K33, K48, and K63) [101,103,104]. Different linkage patterns of ubiquitination can have different functional effects on substrate proteins (Figure 3B). Moreover, ubiquitination is a very dynamic and reversible process. A separate group of enzymes known as deubiquitinases, deubiquitinating enzymes or DUBs, can reverse ubiquitination through enzymatic cleavage of ubiquitin moieties from target substrates. [105,106].

Initially, ubiquitination was discovered as an essential process in quality control of proteins, but lately, diverse processes like cell signaling, protein trafficking, and protein localization were found to be associated with it [107,108]. Ubiquitination is carried out by the sequential action of three different enzymes, (1) ubiquitin-activating enzyme (E1); (2) ubiquitin-conjugating enzyme (E2); and (3) ubiquitin ligase (E3) [109,110] (Figure 3A). Ubiquitin is activated through an ATP-dependent mechanism by the E1 enzyme and subsequently gets transferred through the above mentioned enzyme cascades (Figure 3A). Ubiquitin forms a thioester intermediate with the enzymes before eventually forming a covalent linkage on one or more lysine residues of its substrates. This process continues forming a chain of ubiquitin (the polyubiquitin chain). Ubiquitin itself has seven lysine residues and an N-terminal methionine residue on which other ubiquitin molecules can get attached forming a wide range of typical or atypical ubiquitin chains [94,111]. Depending on the chain topology and structure, different signaling outputs can exist (Figure 3B). Frequently, after attachment of multiple ubiquitin molecules to the target protein, the target substrate gets degraded by a multi-subunit catalytic complex called the proteasome (Figure 3A) [112,113].

## 5. E3 Ligases and Their Involvement in TGFβ Signaling

E3 ubiquitin ligases are primarily responsible for linking the ubiquitin machinery to a specific target substrate protein. Deregulation of E3 ubiquitin ligases has been linked to the development of cancer [114,115]. The E3 ubiquitin ligases can be classified into four major groups based on specific structural motifs. These are (i) HECT domain containing E3 ligase, (ii) really interesting new gene (RING)-finger type, (iii) U-box type, and (iv) plant homeo domain (PHD)-finger type proteins [116]. Distinct ligase domains can have particular ways of ubiquitin transfer. For example, RING domain containing E3 ligases act as a scaffold bringing together an E2 enzyme and its substrate and transfer of ubiquitin takes place directly from E2 to the substrate without formation of an E3-ubiquitin intermediate. In HECT domain containing E3 ligases, however, an E3-ubiquitin intermediate is formed before ubiquitin transfer to its substrates [117].

### 5.1. Role of HECT E3 Ligases in TGFβ Signaling Pathway

Among HECT domain E3 ligases, SMAD ubiquitin regulatory factor 1 (SMURF1) and SMURF2 were shown to have a negative effect on TGFβ/BMP signaling [27,118,119] (Figure 4A). Initially, SMURF1 was identified as a factor that controls steady-state levels of R-SMAD1/5 involved in BMP signaling. SMURF1 induces K-48 linked ubiquitination followed by degradation of SMAD1/5 [118,119]. SMURF1 also negatively regulates BMP signaling through the action of I-SMAD6/7 [120,121]. SMURF1 interacts with nuclear resident SMAD7 and after ubiquitination, it is relocated to the cytoplasm. Additionally, SMURF1 binds to TGF type I receptors (TβRI) in complex with SMAD7, ubiquitinating the receptor followed by degradation of both the receptor and SMAD7 in a proteasome-dependent manner [121]. Thus, ubiquitinated-SMAD7 functions as an adaptor molecule during the degradation of TβRI by recruiting SMURF1 to the receptor (Figure 4A). SMAD4 ubiquitination and stability is also affected by SMURF1, as SMURF1 can bind with SMAD4 in the presence of SMAD6 or SMAD7 and ubiquitinate SMAD4 in the cytoplasm [122,123] (Figure 4B). Indeed, SMAD7 mutants incapable of binding with SMAD4 fail to induce SMURF1-mediated degradation of SMAD4.

SMURF2 can interact with R-SMAD1, 2, and 3, thereby destabilizing TGFβ/BMP signaling [27,124,125]. Its preferred target is SMAD2 as it exhibits higher binding affinity for SMAD2. Although contrasting reports also suggests that in a *Xenopus* model, ectopic SMURF2 specifically inhibits SMAD1 and downregulates BMP signaling [124]. SMURF2 interacts with SMAD2 through its WW domain and proline rich PPXY motif of target SMADs. Like SMURF1, SMURF2 can also enhance the degradation of SMAD4 in complexes with either SMAD7 or SMAD2 (Figure 4B) [122]. SMAD7 associates with SMURF2 in the nucleus and induces export of the complex to activated TβRI. There, it induces K-48 linked poly-ubiquitin chain-mediated degradation of the receptor complex and SMAD7 [122]. Interestingly, studies have shown that SMURF2 can be subjected to functional changes through other post-translational modifications arising from signaling cross-talk with TGFβ and other pathways. Recent studies have highlighted two c-Src kinase phosphorylation sites on SMURF2, Y314 and Y434, which inhibit the interaction between SMURF2 and SMAD7 [126]. Other ubiquitination sites on both SMURF2 and SMAD7 at the WW domain and PPXY motifs have also been well studied [127,128].

Another HECT family member, NEDD4-2 is a well-established binding partner for SMAD7, causing K-48-linked polyubiquitin-mediated degradation of the protein. It can also interact with SMAD2 or SMAD3 and promote ubiquitin-mediated degradation of SMAD2, but not SMAD3 [28]. Like SMURF1/2, Nedd4-2 can also bind to SMAD4 in presence of SMAD7 and induce its ubiquitination and degradation [122] (Figure 4B). Thus, high expression of NEDD4-2 causes transcriptional downregulation of TGFβ/BMP target genes, whereas silencing NEDD4-2 activity elevates the signaling activity and target gene response. Among other HECT ligases, WWP1 (WW domain-containing protein1) or TGIF interacting ubiquitin ligase 1 (TIU1) is a regulator of SMAD7. It associates with SMAD7 and induces its cytosolic localization and binding to TβRI, followed by the K-48 linked polyubiquitination and degradation of the active receptor [129]. Although this affects localization of SMAD7, it does not affect the intracellular stability of SMAD7 [123]. WWP1 also mediates ubiquitination of SMAD4 when in complex with SMAD7, which explains how activation of WWP1 results in diminishing of TGFβ-induced transcriptional responses [122] (Figure 4B). Reports also suggest that WWP1 interacts with SMAD2 and nuclear co-repressor TG interacting factor [TGIF] [130]. This interaction of WWP1 with the nuclear pool of SMAD2 and TGIF leads to degradation of SMAD2, thereby mitigating SMAD2-mediated responses. Thus, knockdown of WWP1 or TGIF leads to stabilization of SMAD2 levels and enhances target gene expression. ITCH is another ligase in this family, known to regulate TGFβ signaling but in a positive way. It facilitates complex formation between SMAD2 and TβR1 and promotes TGFβ-induced transcriptional responses [131]. ITCH promotes SMAD2 phosphorylation, for which the E3 ligase activity of ITCH is needed (Figure 5A). In mouse embryonic fibroblasts, Itch could activate SMAD2 phosphorylation levels but it does not affect overall SMAD2 stability directly [131].

Taken together, it is evident that although a number of HECT E3 ligases affect TGFβ signaling in a negative way, some ligases affect the pathway in a positive way (Figure 4A,B and Figure 5A,B).

### 5.2. Role of RING E3 Ligases in the TGFβ Signaling Pathway

RING E3 ligases affect TGFβ signaling at various points. For simplicity, we will discuss different RING E3 ligases in connection with their target locations in TGFβ signaling.

#### 5.2.1. Targeting Receptors Directly

Tumor necrosis receptor-associated factor (TRAF)6 plays a role in TGFβ-mediated transcriptional regulation during cancer progression. Upon binding of TGFβ with its receptor, TRAF6 causes K-63-linked polyubiquitination of TβRI, which leads to cleavage and release of the TβRI intracellular domain. The intracellular domain associates with transcriptional coactivator p300 and transcriptionally activates tumor associated genes like *SNAIL* and *MMP2* [132] (Figure 5A). TRAF6 is also crucial for TGFβ-mediated activation of JNK and p38 MAP kinase (Figure 5B). Interestingly, TGFβ induces K-63 linked (auto-)ubiquitination of TRAF6 and promotes association between ubiquitinated TRAF6 and TGFβ activated kinase (TAK)1 [133] (Figure 5B). Another TRAF family member, TRAF4 also plays a role in regulating TGFβ signaling. Upon stimulation of cells with TGFβ, TRAF4 is recruited to a TGFβ receptor complex where it recruits USP15, a deubiquitinating enzyme, to the receptor leading to deubiquitination of the receptor and prolonged signaling [134] (Figure 5A). Unlike TRAF6, TRAF4 causes ubiquitination-mediated degradation of SMURF2 and SMURF1 (Figure 5A), which leads to malignancy in certain tumor types [135]. Like TRAF6, TRAF4 also interacts with TβRI, which leads to the K-63-linked polyubiquitination of TRAF4 and activation of TAK1 (Figure 5B).

Cullin ring ligases (CRLs) are the largest family of E3 ligases which can be classified as a subfamily of RING E3 ligases. CRLs function as part of the SCF multiprotein complexes that contains Skp, Cullin, and F-boxes proteins. Recently one CRL, Von Hippel–Lindau ligase (VHL), has been shown to ubiquitinate TβRI. The best-established function of VHL is its role as an E3 ubiquitin ligase for hypoxia-inducible factors (HIFs). In normal oxygen conditions, it ubiquitinates HIF, leading to its degradation. Studies have shown that TGFβ signaling enhances expression of HIF-1α/2α and their target genes even under normoxic conditions, which is dependent on the kinase activity of TβRI and the status of VHL [136]. It was also shown that VHL could ubiquitinate TβRI in a K-48 dependent manner resulting in degradation of the receptor.

#### 5.2.2. Targeting R-SMADs

Among RING domain containing E3 ligases, Casitas B-lineage lymphoma (CBL-b) plays an important role in positively regulating TGFβ signaling in T-cells. CBL-b enhances TGFβ-mediated phosphorylation of SMAD2, increasing signaling output [137] (Figure 5A). Among other RING ligases, Tripartite motif (TRIM)62 negatively regulates TGFβ signaling by binding to MH2 domains of SMAD3, which promotes its ubiquitin-mediated degradation. This results in a decrease in expression of SMAD3 target genes [138]. PRAJA is a RING domain containing E3 ubiquitin ligase, which interacts with embryonic liver fodrin (ELF). ELFs are adapter proteins associated with TGFβ signaling that help recruit SMAD3 to the receptor. PRAJA facilitates ubiquitination of both ELF and SMAD3 and decreases the levels of these proteins, leading to suppression of signaling in a TGFβ-dependent manner [139,140].

ROC1, a RING finger protein enhances degradation of SMAD3. ROC1 forms an SCF ROC1–SCF–βTrCP1 complex consisting of ROC1, Skp1, Cullin1, and β-TrCP1 (also known as Fbw1a) to induce ubiquitination of SMAD3. p300, a nuclear resident transcriptional coactivator interacts with nuclear SMAD3 and facilitates the interaction with ROC1–SCF–βTrCP1. This initiates its export from the nucleus to the cytoplasm for proteolysis [141].

Anaphase promoting complex (APC) has been shown to effectively ubiquitinate and degrade SNoN. SNoN is a negative regulator of TGFβ target genes. TGFβ mediated degradation of SNoN is elicited through APC, which is crucial for amplification of the signal. In this process, SMAD2 and SMAD3 function as adapter proteins in recruiting SNoN and APC in close proximity to each other for subsequent ubiquitination of SNoN by APC [142,143]. RNF111/ARKADIA contains a RING-finger domain in its C-terminus. It interacts with various negative regulators of TGFβ signaling, including c-SKI and SNoN to upregulate TGFβ signaling [144,145]. Some binding partners of ARKADIA have been reported to act as regulators of its function. Studies have shown that the four and a half LIM-only protein 2 (FHL2) interacts with ARKADIA and synergistically cooperates to activate SMAD3/SMAD4-dependent transcription [146]. In addition, FHL2 increases the half-life of ARKADIA through inhibition of its K63 and K27-linked polyubiquitination.

#### 5.2.3. Acting on Co-SMAD

Transcriptional intermediary factor 1 γ (TIF1γ), or TRIM33/RFG7/Ectodermin/PTC7, is a RING family member E3 ubiquitin-ligase which can negatively regulate TGFβ signaling. TIF1γ can induce monoubiquitination of SMAD4, resulting in inhibition of SMAD2/3 complex formation with SMAD4 [147,148]. It can also physically compete with phosphorylated SMAD2/3 for binding with SMAD4 and in this way negatively regulate TGFβ signaling [149]. Skp2, the F-box component of SCF–Skp2, can physically interact with SMAD4. SMAD4 mutants exhibit significantly increased binding affinity to Skp2, which in turn increases their ubiquitination and proteolysis [150]. SCF–βTrCP1 is a critical regulator of SMAD4 stability. Importantly, SCF–βTrCP1 interacts with SMAD4, but not with SMAD2. It also interacts weakly with SMAD3, but this interaction is dependent on SMAD4 (Figure 4B). Increased expression of SCF–βTrCP1 induces the ubiquitination and degradation of SMAD4, while knockdown of βTrCP1 increases SMAD4 expression. Cells that overexpress SCF–βTrCP1 have reduced TGFβ-signaling activity and an impaired cell cycle arrest [123,151].

#### 5.2.4. Regulation of I-SMADs

CBL-b also targets SMAD7 through ubiquitination-mediated degradation, relieving pathway inhibition. CBL-b and SMAD7 interact physically and genetically. It was shown that SMAD7 inactivation restores the TGFβ signaling defect in CBL-b deficient T cells [152]. RNF111/ARKADIA associates with SMAD7 to induce ubiquitination and proteasomal degradation of SMAD7 [153] but it does not bind to TβRI in complex with SMAD7 and fails to induce the degradation of the receptor. RNF12 is a RING domain containing E3 ubiquitin ligase which has been documented to ubiquitinate and degrade SMAD7. Just like Arkadia, RNF12 can enhance TGFβ signaling by specifically degrading an inhibitory SMAD7 [154]. Studies also hint at an indirect mechanism of controlling TGFβ signaling by RNF12. It is documented that RNF12 can interact with SMURF2, which enhances TGFβ responsiveness in U2OS osteosarcoma cells. The dynamics and kinetics of this interaction is poorly understood [155].

All in all, both positive and negative control of the TGFβ pathway is carried out by different RING E3 ligases. RING ligases not only regulate TGFβ signaling by regulating receptor levels or SMAD signaling components, they also affect stability and functioning of HECT E3 ligases like SMURF, or negative regulators of the TGFβ pathway.

### 5.3. Other E3 Ligases

Carboxy-terminus of Hsc70 interacting protein (CHIP) is a U-box-dependent E3 ubiquitin ligase capable of interacting with SMAD1 [156]. Over-expression of CHIP results in ubiquitin-mediated degradation of SMAD1 and SMAD4. Conversely, knockdown of CHIP levels results in heightened BMP signaling [157]. CHIP also can mediate ubiquitination and degradation of SMAD3 to regulate the basal level of SMAD3. Upon ectopic expression of CHIP, SMAD3 is greatly decreased and TGFβ signaling is mitigated. Through quality control of SMAD3, CHIP can modulate the sensitivity of the TGFβ signaling [158].

## 6. E3 Ubiquitin Ligases in Non-Canonical TGFβ Pathways

TGFβ-mediated non-canonical signaling pathways have been well-studied and documented in recent years. The non-canonical pathways are mediated by effector molecules other than SMADs, such as the previously mentioned TAK1-mediated p38 and JNK MAPK pathways. The E3 ubiquitin ligase TRAF6 binds to the receptor which causes its auto-ubiquitination-mediated activation. The activated TRAF6 now binds to TAK1 and activates it in a ubiquitin-dependent manner (Figure 5B). The activated TAK1 now relays the signaling through p38 and JNK [159]. Apart from TGFβ/BMP signaling, TAK1 is able to perform its different biological actions through several other signaling pathways, namely Wnt/Fz, JNK/p38, and nuclear factor (NF)-κB pathways. According to biological context, TAK1 makes complexes with diverse proteins e.g., TAK1-binding proteins (TABs) and TRAFs, which diversifies the role of TAK1. TAK1 plays a crucial role in controlling the activity of IκB kinase (IKK), which in turn activates the transcription factors activator protein (AP)-1 in response to TGF-β/BMP and NF-κB during inflammation [160]. Its noteworthy that SMAD6 can act as a negative regulator of the non-canonical signaling pathway. SMAD6 has been shown to recruit A20, a de-ubiquitinating enzyme that de-ubiquitinates and inactivates TRAF6 and thereby attenuates non-canonical signaling [161]. SMURF1 has also been shown to function downstream of TGFβ as a regulator of RhoA signaling. TGFβ is known to affect the RhoA pathway, and this is important for TGFβ-induced EMT. PAR6 interacts with the TGFβ receptor and is a substrate of TβRII. Phosphorylation of PAR6 is critical for TGFβ-dependent EMT in breast cancer models (Table 1). TGFβ controls the interaction of polarity protein PAR6 with SMURF1. SMURF1, in turn, targets the guanosine triphosphatase RhoA for degradation. Furthermore, SMURF1 was shown to be phosphorylated, thereby switching its substrate specificity from PAR6 to RhoA.

## 7. Role of E3 Ligases Associated with TGFβ Signaling in Mediating Tumorigenesis

To date, mutational analyses, and transcriptional and protein expression data strongly point to a role for E3 ligases in tumorigenesis. Several E3 ubiquitin ligase mutations are known to drive human cancers (e.g., 91% of cases of clear cell renal cell carcinoma (ccRCC) show biallelic *VHL* inactivation, *WWP1* mutation in prostate cancer, deletion in gene enoding parkin RBR E3 ubiquitin protein ligase [PARK2] in ovarian, bladder, and breast cancer) This section summarizes the roles of diverse E3 ligases in perturbing TGFβ signaling activity, ultimately leading to cancer progression.

### 7.1. SMURF1 and SMURF2

The role of SMURF1 has been implicated in a variety of cancers. Genomic hybridization suggests that *SMURF1* is a determining factor for oncogenesis in pancreatic cancer and gastric cancer [166,167]. SMURF1 levels are inversely corelated with survival rate in patients with gastric cancer (GC), colorectal cancer, and ccRCC [168,169]. Knock down of SMURF1 reduces tumorigenesis in a variety of cancer cell models such as of pancreatic, prostate, and ovarian cancer [167,169,170,171]. In addition, SMURF1 is necessary for cancer stem cell maintenance in HNSCC [172]. Elevated levels of SMURF1 were also documented to be associated with thyroid tumor tissues [173]. It is important to mention that the above mentioned roles of SMURF1 in cancer progression are not solely dependent on its effects on TGFβ signaling, but rather due to its impact on diverse signaling events, explained below [118]. PAR6 is a key protein that maintains cell polarity and tight-junction assembly in epithelial cells. As mentioned before, TGFβ controls the interaction between PAR6 and SMURF1, which targets RhoA for degradation, thereby leading to a loss of tight junctions, a hallmark of a mesenchymal transition. SMURF1 is also implicated in bone metastasis in breast cancer models by increasing TGFβ signaling.

SMURF2 acts as a ubiquitinating enzyme for SMURF1 and hence promotes degradation of its sister protein in the early stages of breast cancer. By contrast, in aggressive stages of cancer, for example, in triple negative breast cancer, the SMURF1 level was shown to neutralize the SMURF2 inhibitory effect of tumorigenesis [174]. Although numerous reports implicate the involvement of SMURF1 in cancer progression, very few cases could be explained by its regulation of TGFβ signaling events. Like SMURF1, SMURF2 is also associated with different types of human cancers. SMURF2 expression is upregulated in several cancers like hepatocellular carcinomas (HCC) and colorectal cancer (CRC) [175,176]. Depletion of SMURF2 reduces the migration and invasion of breast carcinomas and CRC [118,177]. Conversely, increased expression of SMURF2 correlates with heightened invasion and lymph node metastases and poor prognosis in some cancers. An inverse correlation between SMURF2 expression and phosphor-SMAD2 levels has also been observed in cancers. In esophageal squamous cell carcinoma, increased expression levels of SMURF2 correlated with tumor development and a poor prognosis [178]. Similar events are apparent in breast and prostate cancers also, suggesting that the repression of TGFβ signaling by SMURF2 occurs during tumor progression [179]. Wu et al. demonstrated the involvement of SMURF2 in progression of pancreatic cancer [180]. SMURF2 was found to promote proliferation, migration, and invasion of pancreatic carcinoma (PANC-1) cells via the TGFβ-induced non-canonical PI3P/AKT pathway [Table 1]. This activation of PI3P signaling suppressed SMAD2/3/FOXO1/PUMA-mediated apoptosis. Upon downregulation of SMURF2, PANC-1 cell invasion and proliferation decreased substantially and apoptosis was induced through FOXO1/PUMA. In lung cancer, SMURF2 was shown to play a cancer-promoting role by ubiquitinating TβR1 [181]. Similar mechanisms and actions of SMURF2 were found to be present in human breast cancer cell lines [182]. There, TGFβ-mediated activation of non-canonical PIP3/AKT/FOXO3a was SMURF2-dependent; upon SMURF2 inactivation, proliferation of breast cancer cells decreased. In summary, mis-regulation of both SMURFs affect TGFβ mediated EMT, migration, and proliferation, while simultaneously inhibiting apoptosis in various cancers.

### 7.2. WWP1

*WWP1* is encoded by the genomic locus, 8q21. A large percentage of breast and prostate cancers show amplification of this locus. Hence, WWP1 is frequently overexpressed in breast and prostate cancers [183,184]. Nguyen Huu et al. found that expression of WWP1 inhibited apoptosis in breast and prostate cancer cells via the inhibition of TGFβ-mediated signaling [185]. In prostate cancer, a mutant of WWP1 was found to be correlate with pathogenesis. The same mutant WWP1^E798V^ displayed increased activity, which ultimately disrupted the TGFβ-induced cytostatic response in the normal, human prostate cell line RWPE [186]. In oral cancer also, WWP1 expression was found to be increased at the mRNA level [187]. In HCC and gastric cancers, expression levels of WWP1 were also found to be increased. Although these studies could not pinpoint the signaling cascade responsible for the cancer-promoting effect, they suggest the mechanisms may involve a negative regulation of TGFβ signaling by WWP1, which may promote cell cycle arrest and induce apoptosis [188,189]. Overall, heightened expression and mutations of WWP1 are the two main causes of their pro-oncogenic functions in various cancers.

### 7.3. NEDD4-2

NEDD4-2 was found to be highly expressed in human distal respiratory epithelium and submucosal glands and ducts and functionally associated with lung cancer development and alveolar fluid regulation [190]. However, it mainly functions through enhancing epithelial sodium channel (EnaC) ubiquitination and cytoplasmic internalization via a direct binding to a highly conserved proline-rich PY motif in the C-termini of αβγ-EnaC, thereby resulting in sodium current decrease and lung edema. Compared with other E3 ubiquitin ligases, NEDD4-2 functions very specifically in the development of lung cancer, whereas its role in promoting TGFβ-mediated effects in other cancer types remain elusive.

### 7.4. TRAFs

TRAF4 can promote both SMAD and non-SMAD signaling downstream of TβRI, during breast cancer progression, and was found to play an important role in mediating TGFβ-induced EMT and metastasis in breast cancer [134]. TRAF4 expression correlated with phosphorylated SMAD2 and phosphorylated TGFβ activated kinase (TAK-1) and poor prognosis in breast cancer patients. TRAF4 is also highly expressed in prostate cancers but there it results in non-proteolytic ubiquitination of receptor tyrosine kinases (RTK’s), which regulates the kinase function [191]. TRAF6 is shown to a play critical role in nuclear accumulation of the intracellular domain of TβR1, which in turn is crucial in EMT and invasion during breast and lung cancer evolution [132]. TRAF6 also plays a critical role during TGFβ-mediated activation of P13K-AKT signaling in prostate cancer [192].

### 7.5. TRIMs

TRIM25 is highly expressed in colorectal and gastric cancer tissues. The ectopic expression of TRIM25 in gastric and colorectal cancer cells activates SMAD and promotes cell migration and invasion [193,194]. TRIM28 facilitates TGFβ-mediated activation of EMT, cell migration, and invasion of lung cancer cells [195]. TRIM59 overexpression has been implicated in several human cancers, such as gastric, lung, and breast cancer [196,197,198]. It is also highly expressed in bladder and breast cancer tissues. TRIM59 promotes the SMAD2/3-mediated signaling response upon TGFβ activation [163,164]. In contrast to TRIM25, TRIM33 acts as a tumor suppressor gene in different cancers. TRIM33 inhibits the invasion and metastasis of both early- and advanced-stage HCC. The expression of TRIM33 is highly reduced in HCC [199] and pancreatic and breast tumors [200,201]. Taken together, these studies show that different TRIM proteins function either as tumor promoters or suppressors.

### 7.6. RNFs

As discussed earlier, RNF12 facilitates degradation of SMAD7, thereby promoting TGFβ signaling activities [202]. RNF12 plays an important role in NR4A1-mediated breast cancer metastasis and invasion. Similar to RNF12, RNF111/ARKADIA, was shown to promote cell migration and metastasis in breast cancer cells but also lung cancer cells [203]. Another TGFβ signaling promoter is RNF38. Expression of RNF38 was found to be upregulated in HCC. RNF38 promoted TGFβ signaling by inducing ubiquitin-mediated degradation of neuroblast differentiation-associated protein (AHNAK), a well-established inhibitor of TGFβ signaling [204].

### 7.7. Other E3 Ligases

Among other E3 ligases, PRAJA was shown to be highly expressed in gastrointestinal cancers [140]. Recent reports also suggested a role for PRAJA in HCC [205]. In pancreatic cancer, SCF–βTrCP1 functions by ubiquitinating and degrading SMAD4 and thereby diminishing TGFβ biological activity. A summary of the activities of different E3 ligases is provided in Table 1. It provides a general overview of their role in TGFβ signaling and progression of cancers.

## 8. E3 Ligases as Emerging Drug Targets

Considering the stimulatory role of TGFβ signaling in cancer progression, different methods have been used to target this pathway to halt cancer cell invasion and metastasis and restore immune responses. While many published reports regarding pre-clinical studies have shown successful responses, the clinical translation towards approved drugs and treatment has so far been unsuccessful [206,207]. The current methods adopted to target TGFβ signaling molecules in clinical trials include antibodies or ligand traps that interfere with ligand-receptor interactions, and small molecule kinase inhibitors. These inhibitors are not selective and also inhibit anti-tumorigenic and homeostatic responses of TGFβ in healthy cells. Systemic inhibition of such general TGFβ targeting agents can elicit toxic side effects that do not outweigh the anti-cancer effects [208]. The dual function of TGFβ in cancer as well as its role in maintaining tissue homeostasis makes it a challenging pathway for drug targeting. As such, selective targeting of positive regulators of pro-oncogenic responses may be an interesting therapeutic strategy to explore. Given the importance of E3 ligases in TGFβ-induced cancer progression, targeting those that are overly active, due to mutation or gene amplification (or overexpressed), in advanced cancer, and those that promote TGFβ signaling in cancer, may work as an alternative, more selective therapeutic strategy.

Various studies have tried to identify E3 ligase inhibitors, with varying degree of success. Cell-based high throughput screening, namely, the ubiquitin reference technique (URT), is one such method that has been used successfully to screen for potent SMURF1 inhibitors capable of blocking SMURF1-mediated degradation of SMAD1 [209], but further studies with these molecules are necessary before they can enter clinical trials. In addition, Chen et al. summarized a wide range of molecules that have been used in various in silico or in vitro studies, with considerable success, against HECT E3 ligases like SMURF1, NEDD4-2, ITCH [210]. It will be interesting to test these molecules in cell-based assays and determine their effect on specific E3 ligases and on TGFβ signaling outputs. Considering the elevated expression of SMURF1, SMURF2, NEDD4-2 in different cancers (Section 7.1 and Section 7.3), similar inhibitors of SMURF2 and NEDD4-2 could be identified. Using a virtual screening tool, Chan et al. successfully identified an inhibitor of the tumor-promoting E3 ligase, SKP2 [211], which plays a role in TGFβ signaling. Similarly, a screening campaign for TRAF6 inhibitors identified molecules that are capable of disrupting TRAF6-Ubc13 interactions and that impede NFκB activation in inflammatory signaling pathways [212]. Mechanistically, the enzymatic pockets of E3 ligases families share high similarity, the selective inhibition of their activities often being achieved by targeting protein–protein interactions (PPIs), e.g., E2-E3 or E3-substrate interactions. The PPIs are frequently mediated by comparatively large contact surface areas with various kinds of interactions playing role. This makes drug development by interfering with PPIs a daunting task [213]. Approaches such as targeted library screening, structure-based drug discovery, or fragment-based drug designing can be applied to target the PPIs surfaces of E3 ligases [214,215,216,217]. Considering the vast array of E3 ligases that have been found to be associated with different cancers, there is an opportunity to develop small molecules that can selectively target E3 ligases and thereby abrogate TGFβ-induced oncogenic responses.

## 9. Emerging Roles of PROTACs and Molecular Glues

In this section, we will focus on two recent therapeutic strategies that can be used for selective targeting of E3 ligases in the future. Both PROTAC (proteolysis targeting chimeras) and molecular glue are novel strategies to target E3 ligases. PROTACs are bi-functional molecules that recruit so-called undruggable targets to their respective E3 ligases. This leads to their accelerated proteolysis and downregulation [218,219]. Some of the PROTACs are presently in clinical trials, potentially highlighting the relevance of this strategy. However, if PROTACs use E3 ligases to degrade other proteins, then how can this strategy be used to inhibit or degrade E3s themselves? The answer lies in numerous reports showing that E3 ligases are capable of auto-ubiquitination, leading to their own degradation. In this regard, a recent study described a new homo-bivalent PROTAC (homo-PROTAC) that simultaneously binds two CRL2^VHL^ molecules and induces their auto-degradation [220]. It should be noted that the role of VHL E3 ligases have been recently linked to TGFβ signaling in a renal cancer model [136]. The highly potent homo-PROTAC CM11 consists of two molecules of the previously described VH298 compound connected via a polyethylene glycol linker [221,222]. The same concept can be applied to other E3 ligases to induce auto-degradation. Similarly, VHL-based PROTACs have been successfully used to degrade SMAD3 proteins in renal carcinoma and renal fibroblast cells [223]. A recent study reported PROTACs, which could degrade TβR1 and inhibit EMT in cancer cells [224]. Another approach developed recently is known as hetero-PROTACs, in which two ligand handles of two PROTACs [VHL and cereblon (CRBN)] were linked together via a linker molecule and tested for the potency of the compound to induce degradation of both ligases. The results showed that VHL could induce degradation of CRBN, but the reverse of this was not observed [225]. Considering the role of different E3 ligases in inducing the degradation of one another, hetero-PROTACs, can be tried against E3 ligases that are implemented in regulating TGFβ signaling.

Molecular glues originally emerged as small molecules that function by inducing gain-of-function interactions between two proteins. Molecular-glue degrons are a subclass of molecule, which facilitate interactions between target proteins and components of the ubiquitin proteasome system to accelerate targeted protein degradation. This strategy holds promise as a unique method for therapeutic molecule discovery for relatively inert protein surfaces that are currently regarded as undruggable. In this context, it can be stated that different E3 ligases are substrates of a competing E3 ligase. For example, βTRCP functions as a ubiquitin ligase for NEDD4, or TRAF4 is ubiquitinated by SMURF1 [226,227]. Therefore, targeting one E3 ligase with another E3 ligase using molecular glue could be an avenue worth exploring.

In vitro screening techniques are relatively easy to design for novel PROTACs and molecular glues, however cell or tissue-based functional assays of the same PROTACs or molecular glues are much harder to design. Solving this is where the challenges lie [212,228]. In the future, identification of diverse chemical moieties with different modes of action for targeting a E3 ligase may serve as a more selective way of inhibiting the tumor promoting activities of TGFβ.

## 10. Conclusions and Perspectives

In this review, we discussed the importance of different E3 ligases in regulating TGFβ signaling. Although the substrate and function of different E3 ligases appear at first glance to be redundant and overlapping, a closer look at the data suggests that is there is a degree of specificity. First, overlapping E3 ligases vary in their cellular distribution and tissue specific expression and secondly the substrates, in some cases, require adaptor proteins for their recruitment of respective E3 ligases. These adapter proteins are critical in recruiting a target protein to diverse E3 ligases. In addition, post-translational modifications play selective roles in recruitment of E3 ligases. In this review, we focused on specific E3 ligases that control TGFβ signaling and whose mis-regulation is correlated with cancer pathogenesis. In diverse cancer databases, correlations exists between genetic driver mutations or transcriptional and translational profiles of E3 ligases, and pathogenesis of different cancers, however, further studies will be needed to precisely delineate the mechanism of action of E3 ligases during pathogenesis. In time, new insights could implicate additional E3 ligases, and layers of complexity, in the regulation of TGFβ canonical and non-canonical signaling. Some E3 ligases have been classified as poor prognosis markers in different cancer models. Hence, targeting those E3s may serve as the path forward in treatment. Considering newer strategies of drug targeting that can specifically target traditionally undruggable proteins, targeting E3 ligases as cancer therapy is potentially within reach.

## Figures and Tables

**Figure 1 ijms-22-00476-f001:**
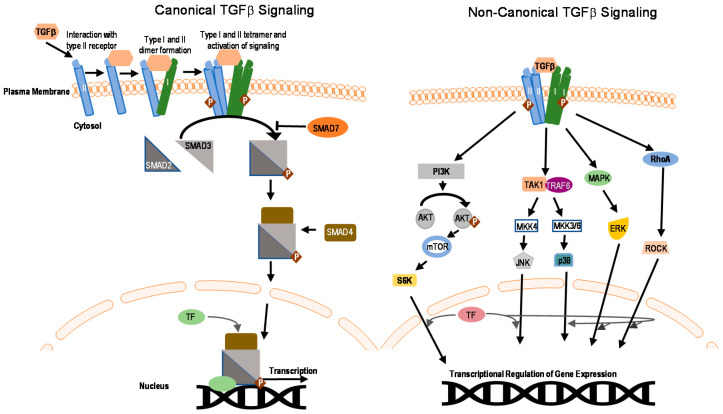
Overview of the transforming growth factor β (TGFβ) signaling pathway. Canonical and non-canonical signaling are shown. TF corresponds to transcription factors. Signaling initiates with binding of TGFβ to the receptors and ends with regulating transcription of target genes. I and II represent TβRI and TβRII, phosphorylation is depicted by P.

**Figure 2 ijms-22-00476-f002:**
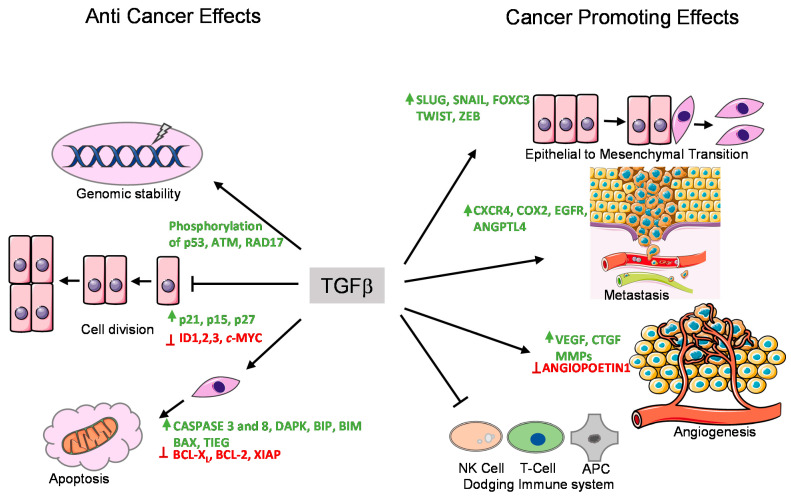
Tumor suppressor and promoting effects of the TGFβ pathway. TGFβ exerts its anti-tumor and tumor promoting effects by regulating the expression of various target genes. These functions are performed by various means shown with arrows. The genes whose expression are enhanced by TGFβ are shown in green, whereas the genes whose expression are suppressed by TGFβ are marked in red.

**Figure 3 ijms-22-00476-f003:**
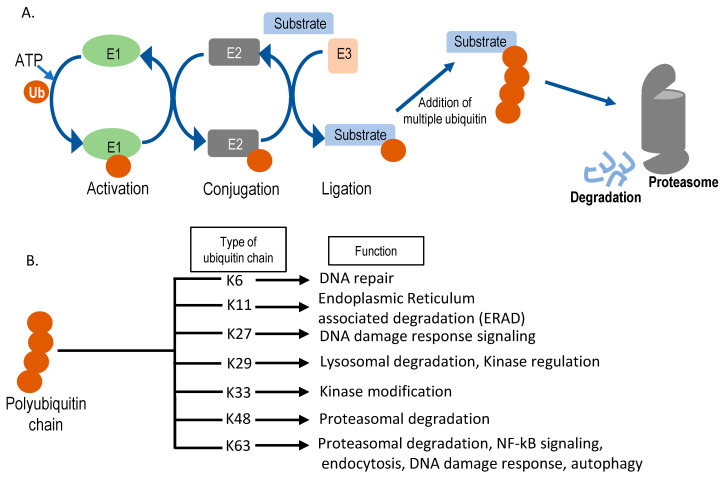
Schematic representation of the ubiquitin-proteasomal degradation pathway. (**A**) Sequential action of E1 (ubiquitin activating enzyme), E2 (ubiquitin conjugating enzyme), and E3 (ubiquitin ligase) enzymes in substrate ubiquitination. Ubiquitin is denoted as Ub. After multiple ubiquitin molecules are attached to the substrate forming lysine (K)-48, it can get degraded by the 26S proteasome. (**B**) Different kinds of polyubiquitin chains formed inside the cell and their cellular functions. Seven different ubiquitin lysine residues can be used for sequential chain formation and, based on the position of chain formation, functions vary.

**Figure 4 ijms-22-00476-f004:**
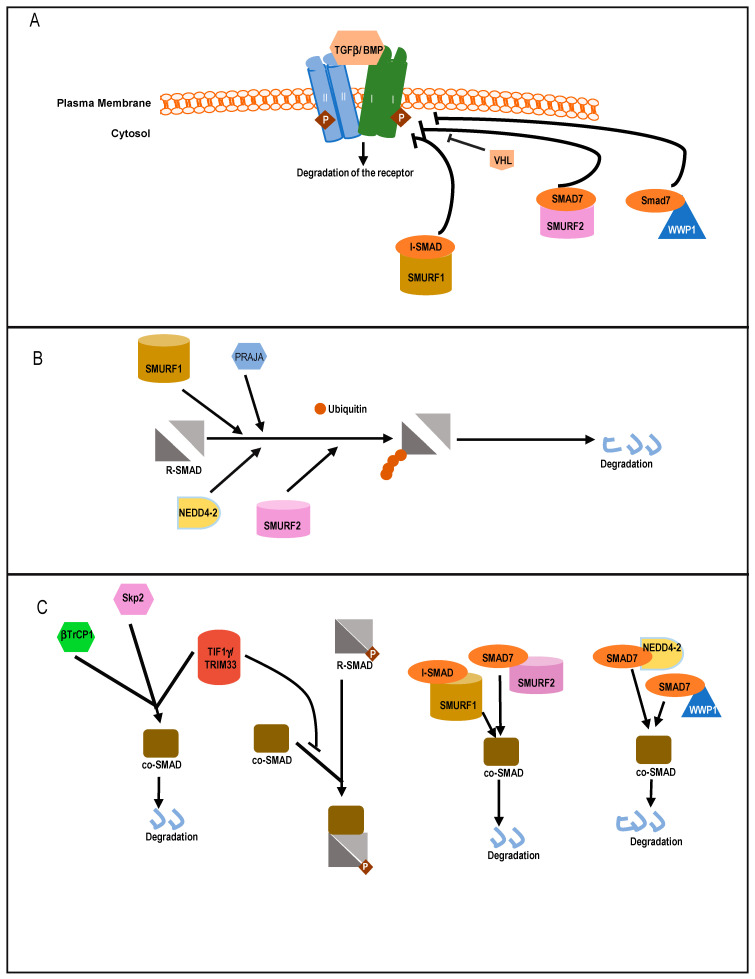
Schematic representation of negative regulation of TGFβ signaling with major E3 ubiquitin ligases. (**A**) Direct inhibition of TβRI by different E3 ligases. SMURF2, WWP1 interact with SMAD7 to inhibit TβRI but SMURF1 can combine with either I-SMADs (SMAD6 or SMAD7) to inhibit TβRI. These E3 ligases induce K-48 mediated polyubiquitination of the receptor, which ultimately induces degradation. (**B**) Inhibition of R-SMAD phosphorylation and stability by diverse E3 ligases. SMURF1 is specific for BMP SMADs (SMAD1/5). PRAJA and NEDD4-2 induce K-48 linked polyubiquitination of SMAD3 and SMAD2, respectively. SMURF2 can ubiquitinate (K-48 linked) R-SMADs (SMAD 1/2 and SMAD3). (**C**) Inhibition and degradation of co-SMAD by different E3 ligases and the role of I-SMADs in this process. SKP2 specifically targets mutant co-SMAD for ubiquitination.

**Figure 5 ijms-22-00476-f005:**
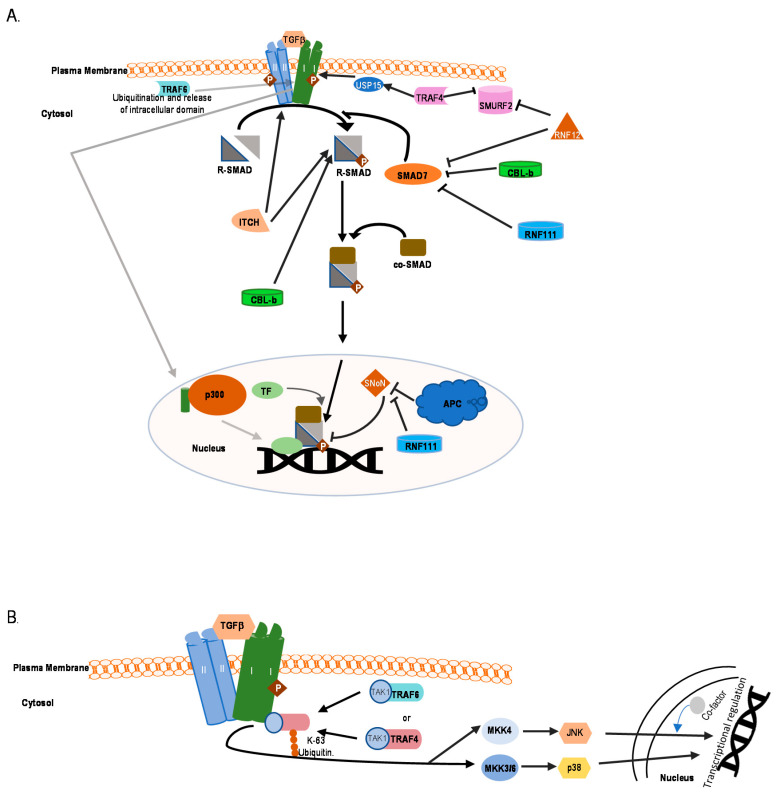
Schematic representation of positive regulation of TGFβ signaling with major E3 ubiquitin ligases and regulation of non-SMAD signaling by TRAF4 or TRAF6 ubiquitin ligases. TβRI and TβRII are indicated as I and II. (**A**) Positive regulation of TGFβ signaling by different E3 ligases. TRAF6 causes ubiquitination and release of the intracellular domain of TβRI, which translocates to the nucleus, associates with p300, and affects transcription of genes. RNF111/ARKADIA inhibits the negative regulator of TGFβ induced transcription SNoN. (**B**) Regulation of JNK/p38 signaling by TRAF4 or TRAF6 E3 ligases.

**Table 1 ijms-22-00476-t001:** List of E3 ubiquitin ligases and their involvement in TGFβ signaling in various cancers.

Broad Group of Ligase	Name	Type of Effect	Target	Effect	Associated Cancer	References
HECT E3 Ligase	SMURF1	Negative	TβRI	Ubiquitination and degradation of receptor	pancreatic cancer, gastric cancer, prostate cancer, ovarian cancer colorectal cancer (CRC),renal cancer, head and neck squamous cell carcinoma (HNSCC), thyroid cancer	[120,121]
			SMAD7	Ubiquitin mediated degradation of SMAD7		[120,121,123]
				Ubiquitination assisted relocation of SMAD7 to cytosol		
			SMAD1/5/8	Ubiquitination and degradation of BMP SMADs		[118,119]
			SMAD4	Turnover of SMAD4 with help of I-SMADs and reducing its overall pool		[122]
	SMURF2	Negative	TβRI/TβRII	Ubiquitination and degradation of receptor complex Ubiquitination assisted relocation and association with TGFβ receptors	breast cancer, hepatocellular carcinoma (HCC), CRC, esophageal squamous cell carcinoma, pancreatic cancer, prostate cancer	[31]
			SMAD7	Degradation of SMAD7		[31,120,121,123]
			SMAD2/3	Induces turnover of SMAD2, thereby preventing SMAD2/3 mediated responses		[124,125]
			SMAD1	Quality control of SMAD1		[124]
			SMAD4	Degradation of Smad4 in complex with I-SMADs		[122]
			SNoN	Ubiquitin mediated degradation of this Transcriptional co-repressor		[162]
	NEDD4-2	Negative	TβRI	Ubiquitination and degradation	lung cancer	[28]
			SMAD2	Ubiquitin mediated turnover of SMAD2		[28]
			SMAD4	Ubiquitination of SMAD4 after SMAD7 assisted interaction		[122]
	WWP1	Negative	TβRI	Downregulation of surface receptor expression through ubiquitin mediated degradation	breast cancer, prostate cancer, HCC, gastric cancer	[129]
			SMAD7	Translocation of SMAD7 to cytosol		[129]
			SMAD4	Ubiquitination of SMAD4 when in complex with SMAD7, leading to its turnover		[122]
			SMAD2	Degradation of nuclear pool of SMAD2		[130]
	ITCH	Positive	SMAD2	Facilitates complex formation between SMAD2 & TβRI by activating SMAD2 phosphorylation		[131]
RING E3 Ligase	CBL-b	Positive	SMAD2	Enhances phosphorylation of SMAD2		[137]
			SMAD7	Ubiquitination and degradation of SMAD7		[152]
	TRIM33	Negative	SMAD4	Inhibition of complex formation with SMAD2/3 via mono-ubiquitination SMAD4	pancreatic cancer, HCC, breast cancer	[147,148]
			SMAD2/3	Bind with SMAD2/3 and reduce association with SMAD4		[149]
	TRIM62	Negative	SMAD3	Ubiquitination and degradation of SMAD3		[138]
	TRIM59	Positive	SMAD2/3	Activate SMAD2/3 mediated signaling upon TGFβ activation	gastric, lung, breast cancer, bladder cancer	[163,164]
	TRAF4	Positive	TβRI/TβRII	Recruitment of USP15 to the receptor to deubiquitinate the receptor	breast cancer, prostate cancer	[134]
			SMURF2	Degradation of SMURF2		[165]
	TRAF6	Positive	TβRI	Site specific cleavage of receptor and transcriptional activation of EMT marker genes	breast cancer, lung, prostate cancer	[133]
	PRAJA	Negative	SMAD3	Ubiquitination of SMAD3 and degradation	gastro-intestinal cancer, HCC	[139]
			ELF	Turnover of ELF		[139,140]
	SCF–SKP2	Negative	SMAD4	Ubiquitin mediated degradation of cancer derived point mutant SMAD4		[150]
	SCF–βTrCP1	Negative	SMAD4	Ubiquitination and degradation of SMAD4	pancreatic cancer	[123,151]
	ROC1	Negative	SMAD3	Ubiquitination of SMAD3 with help from p300, which ultimately leads to SMAD3 degradation in cytosol		[141]
	APC	Positive	SNoN	TGFβ mediated degradation of SNoN		[142,143]
	VHL	Negative	TβRI	Ubiquitination and degradation of TβRI	renal cell carcinoma	[136]
	ARKADIA/RNF111	Positive	SMAD7	Ubiquitin mediated turnover of SMAD7	lung cancer, breast cancer	[153]
			SNoN	Degrades SNoN and stabilizes TGFβ responsive genes		[144,145]
	RNF12	Positive	SMAD7	Ubiquitin mediated degradation of SMAD7	breast cancer	[154]
Other E3 Ligases	CHIP	Negative	SMAD1	Proteasomal degradation of SMAD1		[156]
			SMAD3	Degradation of SMAD3		[158]
			SMAD4	Turnover of SMAD4		[157]

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
