# Peer review of "E3 Ubiquitin Ligases: Key Regulators of TGFβ Signaling in Cancer Progression"

_ijms, 2021, doi:10.3390/ijms22020476_

Round 1

Reviewer 1 Report

The review " E3 ubiquitin ligases: key regulators of TGF signaling in cancer progression” is a review about E3 ubiquitin ligases role in TGFb signaling and discuss potential clinical apllications. In general the review is interesting and well organized but I have some concerns:

Major point:

- This review is focussed in canonical TGF pathway. Authors explain that there are a canonical and no-canonical TGF pathway but after they only explain E3 ubiquitin ligases role in the canonical pathway. In my opinion it would be interesting to add a section about E3 ubiquitin ligases in no canonical TGF pathways.

Minor points:

- The indentation of different paragraphs must ve reviewed.

Author Response

Here our response to reviewer 1 comments.

We are grateful for his/her comments.

The review " E3 ubiquitin ligases: key regulators of TGFβ signaling in cancer progression” is a review about E3 ubiquitin ligases role in TGFβ signaling and discuss potential clinical applications. In general the review is interesting and well organized but I have some concerns:

Major point:

- This review is focused in canonical TGF pathway. Authors explain that there are a canonical and no-canonical TGF pathway but after they only explain E3 ubiquitin ligases role in the canonical pathway. In my opinion it would be interesting to add a section about E3 ubiquitin ligases in no canonical TGF pathways.

Answer: We thank the reviewer for suggesting addition of one section on E3 ubiquitin ligases in non-canonical TGFβ pathways. We have added this, see SECTION 6: E3 ubiquitin ligases in non-canonical TGFβ signaling.

Minor points:

- The indentation of different paragraphs must be reviewed.

 Answer: We have again checked for indentation and they have been corrected.

Reviewer 2 Report

This is a comprehensive review covering the regulation of TGFbeta signalling pathways by ubiquitination, with a specific focus on TGFbeta signalling in cancer. To my knowledge there is no such review in the literature and so this adds a useful resource to the field.

Minor comments:

Line 29, begin with 'Transforming growth factor-beta' rather than 'The transforming growth factor-beta'

Line 41, should be 'effect' instead of 'affect'

Line 100, should read 'such as EMT and apoptosis' rather than 'such EMT and apoptosis'

Line 131, should be 'reversible' not 'reversable'

I suggest that the review would flow more logically by moving section 3 (ubiquitination) and 5 (E3 ligases) beside each other rather than than outlining the control of protein function by ubiquitination, then discussing TGFbeta as a tumour suppressor, followed by the section on E3 ligases.

I suggest that inclusion of a summary table listing the E3s discussed would be useful to indicate

  • whether they have a positive or negative influence on TGFbeta signalling
  • where they act in the pathway
  • whether/how they are dysregulated in cancer

Author Response

Please see our response to reviewer 2 below.

We are grateful for his/her comments. 

We had the manuscript carefully checked by an expert in signal transduction and native English speaker to improve clarity of our manuscript and correct typographoical/garmaticall errors.

This is a comprehensive review covering the regulation of TGFbeta signalling pathways by ubiquitination, with a specific focus on TGFbeta signalling in cancer. To my knowledge there is no such review in the literature and so this adds a useful resource to the field.

Minor comments:

Line 29, begin with 'Transforming growth factor-beta' rather than 'The transforming growth factor-beta'

Line 41, should be 'effect' instead of 'affect'

Line 100, should read 'such as EMT and apoptosis' rather than 'such EMT and apoptosis'

Line 131, should be 'reversible' not 'reversable'

Answer: We thank reviewer for highlighting the above mistakes. We have corrected those in revised version of manuscript.

I suggest that the review would flow more logically by moving section 3 (ubiquitination) and 5 (E3 ligases) beside each other rather than than outlining the control of protein function by ubiquitination, then discussing TGFbeta as a tumour suppressor, followed by the section on E3 ligases.

Answer: We agree with reviewer and corrected these accordingly in the revised manuscript.

I suggest that inclusion of a summary table listing the E3s discussed would be useful to indicate

  • whether they have a positive or negative influence on TGFbeta signalling

Answer: We have included positive or negative effects on signaling in Table1.

  • where they act in the pathway

Answer: In Table1 we have already color coded the signaling molecules that are affected by different E3 ligase. For example: If the E3 ligases work on SMAD2 or 3 it is colored as violet, if the target is SMAD7 it is blue colored, if target is SMAD 4 it is red. Likewise the other targets also are colored with unique colors. Since most of the E3 ligases work on multiple targets in the pathway then one can find multiple color codes for the same ligase conferring its diverse targets. The corresponding references are also highlighted with specific colors.

  • whether/how they are dysregulated in cancer

Answer: TGFβ/SMAD pathway is involved in both tumor suppression and tumor promoting. E3 ligases that regulate this pathway can depending on cellular context can have dual effects. It will be difficult to add these details in a table. Also there would be too much overlap with the main text. We therefore have not indicated whether/how they are dysregulated in cancer, and instead like to refer to the main text.

Round 2

Reviewer 1 Report

Manuscript has been improved and in my opinion it can be published in Int J Mol Sci.